# CXCL10/IP10 as a Biomarker Linking Multisystem Inflammatory Syndrome and Left Ventricular Dysfunction in Children with SARS-CoV-2

**DOI:** 10.3390/jcm11051416

**Published:** 2022-03-04

**Authors:** Eviç Zeynep Başar, Hafize Emine Sönmez, Hüseyin Uzuner, Aynur Karadenizli, Hüseyin Salih Güngör, Gökmen Akgün, Ayşe Filiz Yetimakman, Selim Öncel, Kadir Babaoğlu

**Affiliations:** 1Division of Pediatric Cardiology, Department of Pediatrics and Child Health, Section of Internal Medical Sciences, Faculty of Medicine, Kocaeli University, Kocaeli 41001, Turkey; huseyingungor287@gmail.com (H.S.G.); babaogluk@yahoo.com (K.B.); 2Division of Pediatric Rheumatology, Department of Pediatrics and Child Health, Section of Internal Medical Sciences, Faculty of Medicine, Kocaeli University, Kocaeli 41001, Turkey; eminesonmez@gmail.com; 3Medical Laboratory Techniques Program, Section of Medical Services and Techniques, Kocaeli Vocational School of Health Services, Kocaeli University, Kocaeli 41001, Turkey; huseyin_uzuner@yahoo.com; 4Antibody Research and Production Laboratory, Faculty of Medicine, Kocaeli University, Kocaeli 41001, Turkey; aynuryk2010@yahoo.com; 5Department of Medical Microbiology, Faculty of Medicine, Kocaeli University, Kocaeli 41001, Turkey; 6Pediatric Cardiology Unit, Darıca Farabi Training and Research Hospital, Kocaeli 41700, Turkey; gkmnakgn@gmail.com; 7Division of Pediatric Intensive Care, Department of Pediatrics and Child Health, Section of Internal Medical Sciences, Faculty of Medicine, Kocaeli University, Kocaeli 41001, Turkey; filizyetimakman@hotmail.com; 8Division of Pediatric Infectious Diseases, Department of Pediatrics and Child Health, Section of Internal Medical Sciences, Faculty of Medicine, Kocaeli University, Kocaeli 41001, Turkey; selimoncel@gmail.com

**Keywords:** SARS-CoV-2, CXCL10/IP 10, multisystem hyperinflammatory syndrome, left ventricular dysfunction

## Abstract

Background: To investigate the diagnostic accuracy of CXCL10/IP10 for left ventricular (LV) dysfunction in multisystemic inflammatory syndrome (MIS-C). Methods: This cross-sectional, longitudinal study included 36 patients with MIS-C. Patients were classified as follows: (1) patients presenting with Kawasaki-like features (group I = 11); (2) patients presenting with LV systolic dysfunction (group II = 9); and (3) other presentations (group III = 3). CXCL10/IP10 levels were measured upon admission and on days 3 and 7 of treatment. Results: Twenty patients were male and 16 were female. The median age of patients at diagnosis was 7.5 (1.5–17) years. All patients had a fever lasting for a median of 4 (2–7) days. Ten patients had LV systolic dysfunction. The duration of hospitalization was longer in group II. Lymphocyte and platelet counts were lower, whereas NT-pro-BNP, troponin-I, D-dimer, and CXCL10/IP10 levels were higher in group II. Baseline levels of CXCL10/IP10 were weakly negatively correlated with ejection fraction (r = −0.387, *p* = 0.022). Receiver operator characteristic curve analysis yielded a cutoff value of CXCL10/IP10 to discriminate patients with LV dysfunction was 1839 pg/mL with sensitivity 88% and specificity 68% (Area under curve (AUC) = 0.827, 95% CI 0.682–0.972, *p* = 0.003). Conclusion: Having a good correlation with cardiac function, CXCL10/IP10 is a potential biomarker to predict LV dysfunction in MIS-C patients.

## 1. Introduction

Multisystem inflammatory syndrome in children (MIS-C) is an emerging phenomenon that occurs in children around 4–5 weeks following SARS CoV-2 infection [1,2]. The clinical picture is characterized by fever, rash, and systemic inflammation resulting in multisystemic organ dysfunction. However, the clinical course of the disease is heterogeneous, varying from patient to patient. A review of the data of the presented cases suggests that MIS-C may follow one of the following three common patterns: (1) a persistent febrile illness with elevated biomarkers of inflammation, but no major organ dysfunction; (2) acute myocarditis-like presentation with myocardial dysfunction, shock, and consequent renal or respiratory failure; and (3), a clinical picture very similar to Kawasaki disease (KD), some of which progress to shock requiring vasopressors [3]. Predicting the disease’s progress in the early stage may help the clinician in individualizing treatment and aid in timely initiation of the appropriate therapy.

C-X-C motif chemokine ligand 10/Interferon-γ inducible protein 10 (CXCL10/IP10) is a chemokine secreted from cells that are stimulated with interferon-γ (IFN-γ). CXCL10/IP10 exerts its effects by binding to the cell surface chemokine receptor CXCR3 and acts as a chemoattractant for T cells. Yuan et al. [4] showed an early rise of IFN-γ-stimulated CXCL10/IP10 expression in cardiomyocytes of mice with Coxsackievirus B3-induced myocarditis. Increased levels of CXCL10/IP10 inhibit viral replication at an early stage and improve heart function by protecting cardiomyocytes from damage [4]. Previous studies showed increased CXCL10/IP10 levels following both viral and nonviral myocarditis [5,6], suggesting that it could be a potential biomarker. Researchers have sought biomarkers that may predict the course of the SARS-CoV-2 disease. Loré et al. [7] evaluated 53 potential biomarkers to determine the factors influencing outcome in COVID-19 and they found that CXCL10/IP10 was the best predictive biomarker for outcome in adults with COVID-19. However, the relationship between CXCL10/IP10 levels and disease course in MIS-C patients has not been fully clarified. A study comparing pediatric and adult COVID-19 patients showed increased and similar IFN gene responses in both groups while this antiviral response resolves faster in children [8]. Caldarale et al. [9] demonstrated increased levels of IL-6, CCL2, CXCL8, CXCL9, and CXCL10/IP10 in patients with MIS-C. However, differences in CXCL10/IP10 levels among clinical subgroups have not been investigated. Herein, we aim to evaluate whether CXCL10/IP10 levels change or not according to the clinical course of MIS-C patients and we also compare CXCL10/IP10 levels with those of other inflammatory markers.

## 2. Materials and Methods

### 2.1. Patients and Sample Collection

This cross-sectional, longitudinal study was conducted between May 2020 and October 2021. Patients who were diagnosed with MIS-C according to Centers of Disease Control and Prevention criteria were included (*n* = 42) [1]. Patients whose blood samples were not suitable (hemolytic sample or insufficient sample volume) for analysis were excluded (*n* = 6). Finally, blood samples of 36 patients were evaluated.

Patients were divided into three subgroups: (1) 11 patients with predominantly Kawasaki-like features (group I); (2) 9 patients predominantly with left ventricular (LV) systolic dysfunction (group II); and (3) 16 patients with a persistent febrile illness and elevated biomarkers of inflammation with no major organ dysfunction (group III). Demographic and clinical features were recorded. 

Peripheral blood samples from all patients were collected upon admission and on days 3 and 7 of treatment. Blood samples were centrifuged at 1200× *g* for 10 min at room temperature to collect serum and stored at −80 °C until further analysis. Levels of CXCL10/IP10 (Invitrogen, Frederick, MD 21704, USA) and interleukin (IL)-6 (Siemens, Immulite, Gwynedd LL55 4EL, UK) were measured using ELISA kits according to the manufacturer’s instructions. Complete blood count, blood chemistries, C-reactive protein (CRP), erythrocyte sedimentation rate (ESR), troponin-I, procalcitonin, N-terminal prohormone (NT-pro) brain natriuretic peptide (BNP), D-dimer, and ferritin were tested as part of the patient evaluation. 

All patients underwent echocardiography by the same pediatric cardiologist (Vivid E9, GE Vingmed echocardiograph, General Electric, Horten, Norway) with B-Mode and M-Mode images and pulsed Doppler measurements. Ejection fraction (EF) >55% and fractional shortening (FS) >28% were considered normal. Cardiac dysfunction was defined as decreased EF (<55%) or FS (<28%). Additionally, cardiac involvement, such as coronary artery involvement, valvular disease, and pericarditis, was noted.

Patients were treated according to the severity of the disease [10]. They were classified as mild, moderate, and severe according to the requirement for O_2_ support, vasoactive agents, or the presence of organ dysfunction. All patients were initially treated with intravenous immunoglobulin (IVIG) (2 g/kg) and enoxaparin (1 mg/kg). Corticosteroids (methylprednisolone) were administered to patients with a moderate or severe course. Anakinra was prescribed to severe patients who were refractory to IVIG and steroid therapy. All patients were given acetylsalicylic acid (3–5 mg/kg) for four to eight weeks after discharge. 

Written informed consent was obtained from all patients or their caregivers as appropriate. Ethical approval was obtained from the local ethics committee.

### 2.2. Statistical Analysis

Statistical analysis was performed using IBM SPSS Statistics for Windows, version 20.0 (SPSS, Chicago, IL, USA). The study variables were investigated using visual (histogram and probability plots) and analytic methods (Kolmogorov–Smirnov and Shapiro–Wilk’s tests) to determine the normality of their distribution. Descriptive analyses are presented as frequency and percentage, median and range, as appropriate. The parameters between subgroups were compared by the Kruskal–Wallis test. Spearman rank correlation test was used for correlational analyses of the data. The variables that showed a *p* value of <0.05 in the univariate analysis were tested in multivariate regression analysis for assessment of risk factors. The variance inflation factor (VIF) was used to reduce multicollinearity. The receiver operating characteristic (ROC) curve was used to demonstrate the sensitivity and specificity of CXCL10/IP10 and its optimal cutoff values for predicting LV dysfunction. A *p* value of <0.05 was considered significant and the confidence interval (CI) was 95%.

## 3. Results

### 3.1. Demographic and Clinical Features, Laboratory Findings, and Management

Serum samples of 36 patients with MIS-C were collected upon admission, before treatment, and on treatment days 3 and 7. Twenty patients were male and 16 were female. The median age of patients at diagnosis was 7.7 (1.5–17) years. All patients presented with a fever lasting a median of 4 (2–7) days. The baseline clinical features of patients are summarized in Table 1. Ten patients had LV systolic dysfunction and the median ejection fraction of all patients was 66% (44–79). The frequency of other cardiovascular findings is given in Table 1.

At diagnosis, lymphopenia and thrombocytopenia were detected in 26 and 17 patients, respectively. Levels of CRP and procalcitonin were elevated in all patients, while ESR was elevated in 28 and ferritin in 21 patients. Twenty-three patients had an increased level of pro-BNP and six had elevated troponin-I levels.

The mean duration of hospitalization was nine (3–18) days. All patients were initially treated with IVIG (2 g/kg) and enoxaparin. One patient required a second dose of IVIG.

The patient who received a second dose of IVIG was diagnosed in May 2020. At that time, there was no accepted treatment algorithm. Nine patients received pulsed methylprednisolone for three consecutive days (15–30 mg/kg/day, maximum dose: 1000 mg/day) and then continued with a dosage of 2 mg/kg/day. Twelve patients received 2 mg/kg daily dose of steroid. Six patients were treated with anti-IL-1 (anakinra), concomitantly. Seven patients needed to be admitted to the intensive care unit. All patients were discharged on acetylsalicylic acid prophylaxis (3–5 mg/kg) for four to eight weeks. 

### 3.2. Comparison of Patients According to Subgroups

Patients were divided into three subgroups according to predominant clinical features. Of these, 11 patients were included in group 1 (Kawasaki-like features), nine were classified as group II (cardiac involvement), and the remaining 16 patients were included in group III (other presentations). Demographic and clinical features and treatments of patients are given in Table 2.

Among the patients in group I with Kawasaki-like features, five fulfilled the classification criteria for KD and the remaining six patients met the criteria for incomplete KD. Patients in group II had quite a different phenotype, presenting with predominantly LV dysfunction, and compared to other subgroups, more patients needed intensive care, pulsed steroids, and anakinra treatment. The duration of hospitalization was longer in group II (median: 12 days) than group I (median 7 days) and group III (median 6 days) (*p* = 0.04). Group III included patients who usually presented with predominantly gastrointestinal features (Table 2).

Of the baseline laboratory parameters, lymphocyte and platelet counts were lower in group II whereas NT-pro-BNP, troponin-I, D-dimer, and CXCL10/IP10 levels were higher in group II (Table 3). 

There was a tendency to higher IL-6 levels in group II, but this did not reach statistical significance. Furthermore, CXCL10/IP10 levels were still high and lymphocyte counts were still low both on days 3 and 7 of treatment (Figure 1).

### 3.3. Potential Role of CXCL10/IP10 to Predict Disease Course 

Patients in group II had higher levels of CXCL10/IP10 at diagnosis and on days 3 and 7 of treatment (Figure 1). LV systolic dysfunction was associated with increased levels of CXCL10/IP10. In correlation analysis, baseline levels of CXCL10/IP10 were weakly negatively correlated with the EF value (r = −0.387, *p* = 0.022). CXCL10/IP10 levels were moderately negatively correlated with lymphocyte counts (r = −0.451, *p* = 0.006) and platelet counts (r = −0.462, *p* = 0.005). Furthermore, the strongest (positive) correlation found in this study was between CXCL10/IP10 levels and levels of D-dimer (r = 0.555, *p* < 0.001).

To identify the probable risk factors for cardiac involvement in MIS-C patients, a multiple linear logistic regression model was used to analyze all laboratory parameters. Highly correlated parameters may result in collinearity. To reduce multicollinearity, the variance inflation factor (VIF) was used and parameters with a high VIF were removed from the multivariate regression model. Removing one of the correlated factors usually does not drastically reduce the R-squared because these variables supply redundant information. This analysis found that increased levels of CXCL10/IP10 (*p* = 0.002) were found to be associated with LV dysfunction in MIS-C patients. The predictive value of CXCL10/IP10 to discriminate patients with LV dysfunction from others was analyzed using ROC curve analysis. The optimal cutoff value for baseline CXCL10/IP10 level was 1839 pg/mL with a sensitivity 82%, specificity 68%, and area under curve (AUC) =0.827, CI 0.682–0.972 (*p* = 0.003); see Figure 2. 

Five patients on day 3 and four patients on day 7 had levels of CXCL10/IP10 that were higher than 1839 pg/mL and their LV functions were still slightly decreased. The remaining patients had normal LV function.

## 4. Discussion

In the present study, the patients who had predominantly cardiac dysfunction presented with a more severe phenotype and had a greater need for intensive care, pulsed steroids, and anakinra than the other subgroups. Furthermore, the correlation between cardiac function and CXCL10/IP10 suggested that baseline CXCL10/IP10 is a candidate biomarker to predict LV dysfunction in MIS-C patients.

The COVID-19 pandemic has resulted in significant morbidity and mortality during the last two years. However, the disease course is quite different in children. Instead of displaying direct respiratory effects of the virus, severe pediatric cases tend to present with a hyperinflammatory response that is now called MIS-C. A recent meta-analysis investigating the predictive factors for poor prognosis in 79,104 pediatric patients with COVID-19 confirmed that patients with MIS-C had an increased risk for mortality (odds ratio (OR) = 58.00, 95% CI 6.39–526.79) [11]. Although initial studies reported MIS-C as a Kawasaki-like disease, it is now accepted that MIS-C may present with different phenotypes and outcomes [3,12,13]. The severity of the disease varies in each patient. According to previous studies, severe MIS-C cases required more frequent intensive care admission due to shock, LV dysfunction, or requirement for inotrope support [3,12,13]. Thus, it is believed that predicting the disease course at diagnosis may help to individualize and optimize the management. For this purpose, studies have focused on identifying potential biomarkers capable of aiding in predicting the disease course. According to a recent meta-analysis, higher white blood cell (WBC) and absolute neutrophil counts, higher levels of CRP, D-dimer, ferritin, and lower absolute lymphocyte counts were associated with a severe course of MIS-C [14]. Furthermore, some parameters, such as osteopontin and endothelial glycocalyx degradation, have been suggested as biomarkers for distinguishing severe COVID-19 and MIS-C patients from mild/asymptomatic children with COVID-19 [15,16]. Kavurt et al. [17] evaluated the correlation between cardiac involvement and laboratory parameters and found that LV systolic dysfunction was associated with increased levels of troponin-I, NT-pro BNP, procalcitonin, and ferritin. Chang et al. [18] reported a correlation between impaired global longitudinal strain values and increased levels of IL-6 and IL-8. However, these studies did not provide a clear cutoff to identify severe cases or LV dysfunction at diagnosis. Herein we showed that baseline levels of CXCL10/IP10 levels were higher in patients who had LV dysfunction and negatively correlated with the extent of EF (r = −0.387, *p* = 0.022). In addition, ROC analysis revealed that the optimal cutoff value for CXCL10/IP10 was 1839 pg/mL with a sensitivity of 82% and specificity of 68%. Five patients on day 3 and four patients on day 7 had CXCL10/IP10 levels higher than 1839 pg/mL and their LV function measurements remained impaired.

CXCL10/IP10 is a chemoattractant chemokine for T cells, exerting its effect through interaction with the cell surface chemokine receptor CXCR3. CXCL10/IP10 is secreted from cells as a response to increased IFN-γ. The role of this chemokine in cardiac diseases has been evaluated previously. For instance, the expression of CXCL10/IP10 by endothelial cells, smooth muscle cells, and macrophages was demonstrated during the formation of atherosclerotic lesions [19,20]. Elevated levels of CXCL10/IP10 were also shown in sera of patients with coronary artery disease [21,22]. However, the exact prognostic implications are not clear yet. CXCL10/IP10 was hypothesized to be a potential biomarker for myocarditis because of studies reporting the expression of CXCL10/IP10 in the heart following viral and nonviral infections [4,5]. Furthermore, CXCL10/IP10 levels were also evaluated in other pediatric acute inflammatory conditions. For instance, Ko et al. [23] evaluated the plasma cytokine profile of patients with KD and found notable IP-10 levels to be present. They concluded that it was a useful marker for the diagnosis of KD. A recent study showed a significant association between genetic polymorphisms of IP10 and the risk of KD [24]. Systemic juvenile idiopathic arthritis (sJIA) is another inflammatory disease sharing similar clinical findings with MIS-C. Increased levels of IFNγ-induced chemokines such as CXCL9, CXCL10, and CXCL11 were found to be associated with sJIA complicating macrophage activation syndrome [25]. Now, a trial of anti-IFNγ monoclonal antibody (emapalumab) in sJIA is ongoing [26]. Besides these inflammatory conditions, CXCL10/IP10 levels were evaluated among adult patients with COVID-19 and were found to be a good predictor of the outcome [7,27]. However, its predictive accuracy has not been elucidated in MIS-C patients as yet. Caldarale et al. [9] compared the inflammatory responses in 10 children with COVID-19 and 9 children with MIS-C. They showed that IL-6, CCL2, CXCL8, CXCL9, and CXCL10/IP10 levels were higher in patients with MIS-C than in those with COVID-19. They concluded that interferon response plays a central role in the pathogenesis of MIS-C patients. However, although increased CXCL10/IP10 levels were demonstrated in MIS-C patients, its impact on the clinical course was not investigated. Herein we sought to investigate if CXCL10/IP10 may be a potential marker to anticipate LV dysfunction. Previous studies have suggested that increased levels of CXCL10/IP10 were associated with nonischemic heart failure and that CXCL10/IP10 was secreted as a result of enhanced Th1 lymphocyte polarization and infiltration into the myocardium [28,29].

D-dimer is a fibrin degradation product and is elevated in the presence of thrombus and/or in the resolution phase of thrombus formation. It is well known that SARS CoV-2 infection predisposes to coagulopathy [30] and endothelial injury is thought to be the underlying factor in disease severity and coagulopathy in patients with COVID-19. Adult studies have confirmed an association between elevated D-dimer levels and the severity of the course of COVID-19 [31,32]. In the present study, there was a significant correlation between D-dimer levels and CXCL10/IP10, which are both elevated in patients with LV dysfunction. These findings support the hypothesis that both endothelial injury and abnormal inflammatory response may be responsible for the pathogenesis of MIS-C.

Caldarale et al. also reported that CXCL10/IP10 levels returned to normal by day 5 of the disease [9]. In the present study, patients with cardiac disease had higher levels of CXCL10/IP10 at admission, day 3, and, interestingly, day 7. Patients in group II required more intensive treatment including pulsed steroids and anakinra. This may be related to prolonged IFN response in these patients. Ouldali et al. [33] showed a more favorable outcome in MIS-C patients treated with IVIG and methylprednisolone than in those who received IVIG alone. They suggested that treatment with IVIG and methylprednisolone reduced the risk of acute ventricular dysfunction and requirement for hemodynamic support. According to our findings, we suggest that CXCL10/IP10 is a potential biomarker for cardiac involvement and the need for more intensive treatment, but these findings require a larger evidence base. Anti-IFNγ monoclonal antibody (emapalumab) may be an alternative treatment in MIS-C patients presenting with LV systolic dysfunction.

Our study was limited by its single-center design and small sample size. Furthermore, the clinical heterogeneity of the patient cohort may be due to successive waves of COVID-19 with variable severity as new SARS-Cov-2 variants emerged. The absence of a healthy control group is another limitation of this study. However, given the relatively low numbers of other studies reporting MIS-C, this report has added to the evidence base for the role of baseline CXCL10/IP10 measurement, where available, as a new biomarker for the prediction of LV dysfunction.

## 5. Conclusions

In conclusion, we suggest that CXCL10/IP10 plays a role in the pathophysiology of MIS-C related to SARS-COV2. Evaluating the accuracy and utility of CXCL10/IP10 in larger prospective studies may help clarify this hypothesis and the exact etiopathology of MIS-C. If CXCL10/IP10 has a central role in this, then novel targeted therapies may emerge, such as anti-IFNγ monoclonal antibodies.

## Figures and Tables

**Figure 1 jcm-11-01416-f001:**
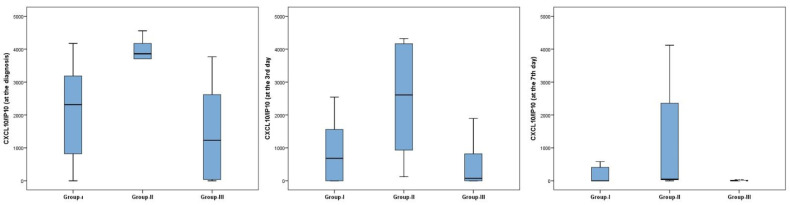
CXCL10/IP10 levels according to subgroups.

**Figure 2 jcm-11-01416-f002:**
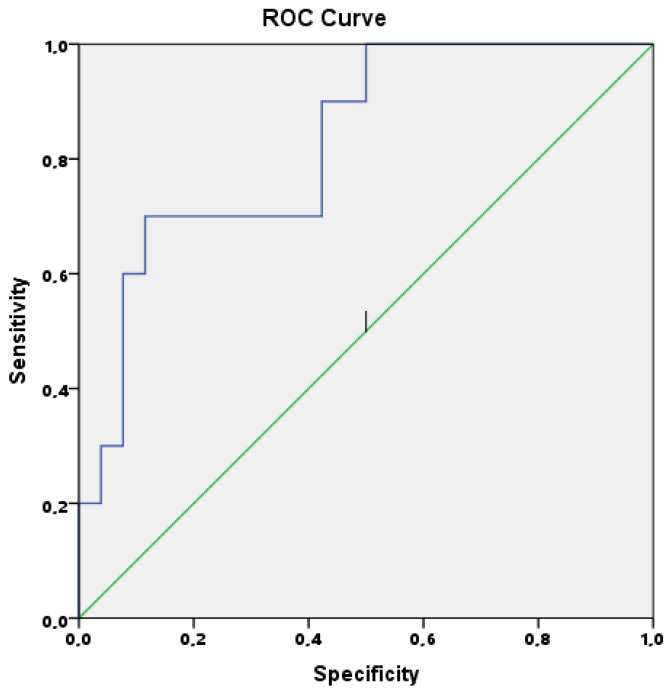
Receiver operating characteristic curve of baseline serum level of CXCL10/IP10.

**Table 1 jcm-11-01416-t001:** Clinical characteristics of all patients.

	Patients (*n* = 36)
Fever, *n*	36
Mucocutaneous findings	
Polymorphous rash, *n*	22
Conjunctivitis, *n*	19
Oral changes, *n*	12
Extremity changes, *n*	8
Cervical lymphadenopathy, *n*	1
Musculoskeletal findings	
Myalgia, *n*	4
Gastrointestinal findings	
Abdominal pain, *n*	21
Diarrhea, *n*	13
Appendicitis or bowel edema, *n*	5
Cardiovascular findings	
LV dysfunction or myocarditis, *n*	10
Pericarditis, *n*	6
Coronary artery dilatation, *n*	1
Coronary artery brightness, *n*	2
Mild mitral valve insufficiency, *n*	13
Moderate-severe mitral valve insufficiency, *n*	4
Mild tricuspid valve insufficiency, *n*	1
Renal involvement, *n*	3
Neurologic involvement, *n*	2

**Table 2 jcm-11-01416-t002:** Comparison of demographic and clinical features and treatments of patients.

	Group I (*n* = 11)	Group II (*n* = 9)	Group III (*n* = 16)	*p*
Gender (Female/Male)	5/6	4/5	7/9	0.996
Fever, *n*	11	9	16	-
Polymorphous rash, *n*	10	6	6	0.009
Conjunctivitis, *n*	8	4	7	0.187
Oral changes, *n*	7	2	3	0.02
Extremity changes, *n*	4	3	1	0.113
Cervical lymphadenopathy, *n*	1	0	0	0.521
Myalgia, *n*	1	2	1	0.813
Abdominal pain, *n*	4	6	11	0.206
Diarrhea, *n*	1	2	10	0.01
Appendicitis or bowel edema, *n*	0	0	5	0.02
LV * dysfunction or myocarditis, *n*	1	9	0	<0.001
Renal involvement, *n*	1	2	0	0.356
Neurologic involvement, *n*	0	2	0	0.139
Intensive care unit, *n*	1	6	0	<0.001
IVIG *, *n*	11	9	16	-
Second dose of IVIG	0	1	0	0.253
Pulse steroid, *n*	2	6	1	0.02
Methylprednisolone (2 mg/kg), *n*	3	2	7	0.02
Anakinra, *n*	1	5	0	0.002

* IVIG, intravenous immunoglobulin; LV, left ventricular.

**Table 3 jcm-11-01416-t003:** Comparison of laboratory findings of patients.

	Group I (*n* = 11)	Group II (*n* = 9)	Group III (*n* = 16)	*p*
Complete Blood Count **				
WBC *, mm^3^ (at diagnosis)	8611 (4328–19,000)	11,666 (1388–18,930)	8341 (3090–23,000)	0.389
WBC *, mm^3^ (at day 3)	13,045 (4471–17,800)	19,200 (1749–28,200)	5713 (3448–9706)	0.015
WBC *, mm^3^ (at day 7)	15,854 (5040–30,220)	11,805 (5061–20,020)	10,641 (4370–20,656)	0.876
Lymphocyte, mm^3^ (at diagnosis)	1801 (604–5190)	451 (248–1080)	1085 (710–3283)	0.002
Lymphocyte, mm^3^ (at day 3)	2993 (1191–7410)	880 (562–1303)	1540 (766–3623)	0.006
Lymphocyte, mm^3^ (at day 7)	3370 (1440–14,420)	2092 (942–3160)	3412 (1640–7061)	0.011
Hemoglobin, g/dL (at diagnosis)	10.9 (8.2–12.5)	10.9 (10.5–10.4)	11.4 (7.7–13.7)	0.641
Hemoglobin, g/dL (at day 3)	10.6 (8.7–12.5)	10 (8.7–14.1)	10.6 (8.8–12.8)	0.897
Hemoglobin, g/dL (at day 7)	11.8 (7.30–13.4)	12 (11–14.3)	11.6 (9.1–13.6)	0.377
Platelet, mm^3^ (at diagnosis)	176,250 (110,000–462,000)	120,500 (55,400–209,000)	195,000 (67,000–384,000)	0.090
Platelet, mm^3^ (at day 3)	341,000 (198,000–665,400)	173,500 (73,600–284,000)	195,500 (180,000–767,000)	0.013
Platelet, mm^3^ (at day 7)	535,000 (412,000–804,000)	340,000 (175,000–410,000)	334,500 (278,000–718,000)	0.250
Inflammatory Markers **				
CXCL10/IP10 *, pg/mL (at diagnosis)	2280 (0–4174)	3938 (1571–4558)	763 (0–346)	0.004
CXCL10/IP10 *, pg/mL (at day 3)	933 (0–2545)	3467 (123–4319)	37 (0–4132)	0.019
CXCL10/IP10 *, pg/mL (at day 7)	116 (0–3161)	2264 (0–4121)	0 (0–577)	0.021
IL-6 *, pg/mL (at diagnosis)	36.4 (3.1–136)	324 (9.1–2330)	18 (0–346)	0.066
IL-6 *, pg/mL (at day 3)	2.6 (0–58)	4.6 (3.3–147)	5.5 (0–22)	0.979
IL-6 *, pg/mL (at day 7)	1.1 (0–24)	0 (0–6)	0 (0–4.5)	0.235
CRP *, mg/L (at diagnosis)	145 (24–333)	187.5 (29–278)	138.5 (1.98–233)	0.388
CRP *, mg/L (at day 3)	33 (8–98)	122.5 (8.9–219)	47 (5.2–109)	0.074
CRP *, mg/L (at day 7)	10.8 (0.94–20.6)	14.8 (2.29–35)	4.7 (2–98)	0.550
ESR *, mm/hr (at diagnosis)	50 (10–115)	42 (4–69)	46 (8–114)	0.421
ESR *, mm/hr (at day 3)	79 (8–140)	24 (13–45)	58 (38–137)	0.027
ESR *, mm/hr (at day 7)	43 (5–140)	13.5 (4–40)	38 (14–63)	0.003
Procalcitonin, ng/mL (at diagnosis)	3.5 (0.48–42)	18.5 (0.61–100)	1.9 (0.24–25.2)	0.070
Procalcitonin, ng/mL (at day 3)	0.25 (0.08–6.2)	1.5 (0.12–88)	0.64 (0.07–16)	0.095
Procalcitonin, ng/mL (at day 7)	0.24 (0.04–1.36)	0.5 (0.04–15)	0.06 (0.02–9.2)	0.545
Ferritin, ug/L (at diagnosis)	337 (142–8632)	616 (127–1746)	141 (12–2275)	0.064
Ferritin, ug/L (at day 3)	384 (135–4103)	723 (279–1889)	195 (48–423)	0.004
Ferritin, ug/L (at day 7)	200 (110–2000)	400 (127–1021)	97 (39–742)	0.018
Cardiac Markers **				
NT-pro-BNP *, ng/L (at diagnosis)	2201 (70–14,900)	2885 (282–24,000)	426 (70–14,100)	0.032
NT-pro-BNP *, ng/L (at day 3)	1083 (106–3960)	2870 (428–35,000)	601 (70–1230)	0.004
NT-pro-BNP *, ng/L (at day 7)	190 (70–894)	181 (123–4150)	70 (70–380)	0.088
Troponin-I, ng/L (at diagnosis)	0 (0–65)	17.5 (0–140)	0 (0–0)	0.006
Troponin-I, ng/L (at day 3)	0 (0–21)	44.5 (0–390)	0 (0–18)	0.097
Troponin-I, ng/L (at day 7)	0 (0–0)	0 (0–39)	0 (0–0)	0.051
Coagulation Parameters **				
D-dimer, µg/mL (at diagnosis)	2.1 (0.6–7.45)	12.8 (1.1–21.8)	1.7 (0.19–12.5)	0.030
D-dimer, µg/mL (at day 3)	1.4 (0.62–2.82)	4.2 (0.85–6.24)	1.07 (0.36–6.5)	0.184
D-dimer, µg/mL (at day 7)	1.22 (0.58–9.87)	2.05 (0.73–384)	0.91 (0.20–11.1)	0.556
Fibrinogen, g/L (at diagnosis)	5.2 (2.6–8.6)	5.1 (2.4–7.9)	4.7 (2.9–6.6)	0.480
Fibrinogen, g/L (at day 3)	4.1 (1.9–8.6)	4.05 (3.23–6.34)	4.3 (2.9–5.73)	0.824
Fibrinogen, g/L (at day 7)	3.5 (0–5.9)	2.3 (1.05–3.91)	3.1 (0.94–4.6)	0.180

* CRP, C-reactive protein; CXCL10/IP10, C-X-C motif chemokine ligand 10/Interferon-γ inducible protein 10; ESR, erythrocyte sedimentation rate; IL-6, interleukin-6; NT-pro-BNP, N-terminal prohormone brain natriuretic peptide; WBC, White blood count; ** Data expressed as median (minimum-maximum).

## Data Availability

The data presented in this study are available on request from the corresponding author.

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
