# Peer review of "CXCL10/IP10 as a Biomarker Linking Multisystem Inflammatory Syndrome and Left Ventricular Dysfunction in Children with SARS-CoV-2"

_jcm, 2022, doi:10.3390/jcm11051416_

Round 1
Reviewer 1 Report
Seeking for new biomarkers in MIS-C/PIMS seems to be the new perspective in clinical research in pediatric cardiology. The authors had very interesting scientific idea which resulted in some preliminary results, but in my view they are not presented in an appropriate way. The authors underline the negative correlation between CXCL10/IP10 baseline level and ejection fraction as the main finding of the study. However, correlation with r=-0.387 has to be considered as weak (not good, as written in line 42). Therefore the conclusions seem to me to be overdrawn. The specificity of the cutoff level according to the ROC curve is very low (50%) and may result in clinical insignificance of CXCL10/IP10 measurements. Can the authors propose other cutoff levels with better specificity or at least discuss about differential diagnosis?
Nevertheless, there is another very interesting correlation with the highest r=0.555 in the whole presented analysis: CXCL10/IP10 vs. D-dimer (line 190). I would like the authors to elaborate on this finding.
When it comes to the 'Results' section, I suggest the authors remove percentage values, as the study group involves only 36 patients and from statistical point of view it is not enough to calculate percentage (which is derived from Latin "per centum" = "one part per hundred"). Are the results shown as median with 95% CI? I assume it from lines 135, 136, but it is not clearly defined in 'Materials and methods' section. If yes, 95% CI of median age is very broad - I suggest recalculating it from months to years.
It is unfortunate to write in line 147 that one patient required a second dose of IVIG. According to the international treatment standards, ineffective IVIG in patients diagnosed with MIS-C/PIMS should be followed by GCS and later with biological agents. From my standpoint, the sentence needs to be rephrased.
The 'Discussion' section is the weakest part of the current version of the manuscript and needs to be heavily edited. It includes plenty of information more adequate for the 'Background' (e.g. lines 243-253). I suggest expanding the 'Discussion' section with summarization of CXCL10/IP10 levels in other pediatric acute inflammatory conditions (e.g. pneumonia, macrophage activation syndrome, Kawasaki disease) in order to address the very important question regarding potential utilization of CXCL10/IP10 in differential diagnosis.
Furthermore, the study limitations are not discussed properly. How about the subjectivity of ultrasound assessment? Have all patients been assessed by the same pediatric cardiologist or maybe two different physicians? If yes, did they perform a cross-check of their results? The study group is not only a small one, above all it is a heterogenous group which has been further subdivided. Moreover, the differences in clinical presentation of MIS-C/PIMS between subsequent COVID-19 waves impede drawing consistent conclusions.
Additionally, the article is seriously linguistically flawed and requires thorough review by a native speaker. Many sentences have inappropriate syntax, probably adapted from other language. Furthermore, here are just a few examples of mistakes: "Increased levels of CXCL10/IP10 was" (line 36), "predictor potential" (line 72), "compare its power with those" (line 79).
I do believe that the revised version of the manuscript may enlighten future readers with its originality. I am looking forward to reading this paper after implementing of all suggested corrections.
Author Response
Comment 1: Seeking new biomarkers in MIS-C/PIMS seems to be the new perspective in clinical research in pediatric cardiology. The authors had a very interesting scientific idea which resulted in some preliminary results, but in my view, they are not presented in an appropriate way. The authors underline the negative correlation between CXCL10/IP10 baseline level and ejection fraction as the main finding of the study. However, correlation with r=-0.387 has to be considered as weak (not good, as written in line 42). Therefore the conclusions seem to me to be overdrawn. The specificity of the cutoff level according to the ROC curve is very low (50%) and may result in clinical insignificance of CXCL10/IP10 measurements. Can the authors propose other cutoff levels with better specificity or at least discuss differential diagnosis?
Response 1: Thank you for your excellent suggestions and recommendations. We hope the revised manuscript has addressed all the concerns that you highlighted.
We agree with you concerning the strength of association between CXCL10/IP10 and ejection fraction and have revised the text that originally occupied line 42, as follows:
“Baseline levels of CXCL10/IP10 were weakly negatively correlated with ejection fraction (r= -0.387, p=0.022)”
We have identified more sensitive and specific cut-off values for CXCL10/IP10 and LV dysfunction and have added this to the Abstract and the Results section:
(Abstract) "ROC curve analysis yielded a cutoff value of CXCL10/IP10 to discriminate patients with LV dysfunction of 1,839 pg/mL with sensitivity 88% and specificity 68% (Area under curve [AUC]=0.827, 95% CI 0.682–0.972, p=0.003).
(Results) "In correlation analysis, baseline levels of CXCL10/IP10 were weakly negatively correlated with the EF value (r= -0.387, p=0.022). CXCL10/IP10 levels were moderately negatively correlated with lymphocyte counts (r= - 0.451, p=0.006) and platelet counts (r= -0.462, p=0.005). Furthermore, the strongest (positive) correlation found in this study was between CXCL10/IP10 levels and levels of D-dimer (r=0.555, p<0.001)."
Furthermore, we have revised the conclusion as follows:
“In conclusion, we suggest that CXCL10/IP10 plays a role in the pathophysiology of MIS-C related to SARS-COV2. Evaluating the accuracy and utility of CXCL10/IP10 in larger prospective studies may help clarify this hypothesis and the exact etiopathology of MIS-C. If CXCL10/IP10 has a central role in this, then novel targeted therapies may emerge, such as anti-IFNγ monoclonal antibodies..”
Comment 2: Nevertheless, there is another very interesting correlation with the highest r=0.555 in the whole presented analysis: CXCL10/IP10 vs. D-dimer (line 190). I would like the authors to elaborate on this finding.
Response 2: Thank you for this apposite observation. We have highlighted this in the revised text of the Results section (see response above).
We have discussed this result as follows:
“D-dimer is a fibrin degradation product and is elevated in the presence of thrombus and/or in the resolution phase of thrombus formation. It is well known that SARS CoV-2 infection predisposes to coagulopathy [30] and endothelial injury is thought to be the underlying factor in disease severity and coagulopathy in patients with COVID-19. Adult studies have confirmed an association between elevated D-dimer levels and the severity of the course of COVID-19 [31, 32]. In the present study, there was a significant correlation between D-dimer levels and CXCL10/IP10, which are both elevated in patients with LV dysfunction. These findings support the hypothesis that both endothelial injury and abnormal inflammatory response may be responsible for the pathogenesis of MIS-C.”
Comment 3: When it comes to the 'Results' section, I suggest the authors remove percentage values, as the study group involves only 36 patients and from the statistical point of view it is not enough to calculate the percentage (which is derived from Latin "per centum" = "one part per hundred"). Are the results shown as median with 95% CI? I assume it from lines 135, 136, but it is not clearly defined in the 'Materials and methods’ section. If yes, 95% CI of median age is very broad - I suggest recalculating it from months to years.
Response 3: Once again, thank you. We have removed the percentage values although they are useful for indicating relative proportions of totals, even if the total is below one hundred. We have also recalculated the ages in decimal years and patients were included ranging in age from 1.5 to 17 years, so yes, the range was broad.
Comment 4: It is unfortunate to write in line 147 that one patient required a second dose of IVIG. According to the international treatment standards, ineffective IVIG in patients diagnosed with MIS-C/PIMS should be followed by GCS and later with biological agents. From my standpoint, the sentence needs to be rephrased.
Response 4: This patient was diagnosed in May 2020, only two months after the first cases were reported in Turkey. At that time, there was no accepted treatment algorithm and the managing clinicians opted to try a second dose of IVIG. We have clarified this issue in the result section, with the following:
"One patient required a second dose of IVIG. The patient who received a second dose of IVIG was diagnosed in May 2020. At that time, there was no accepted treatment algorithm."
Comment 5: The 'Discussion' section is the weakest part of the current version of the manuscript and needs to be heavily edited. It includes plenty of information more adequate for the 'Background' (e.g. lines 243-253). I suggest expanding the 'Discussion' section with the summarization of CXCL10/IP10 levels in other pediatric acute inflammatory conditions (e.g. pneumonia, macrophage activation syndrome, Kawasaki disease) in order to address the very important question regarding potential utilization of CXCL10/IP10 in the differential diagnosis.
Response 5: We thank Reviewer 1 for this advice and have revised the Discussion accordingly
“Furthermore, CXCL10/IP10 levels were also evaluated in other pediatric acute inflammatory conditions. For instance, Ko et al. [23] evaluated the plasma cytokine profile of patients with KD and found notable IP-10 levels to be present. They concluded that it was a useful marker for the diagnosis of KD. A recent study showed a significant association between genetic polymorphisms of IP10 and the risk of KD [24]. Systemic juvenile idiopathic arthritis (sJIA) is another inflammatory disease sharing similar clinical findings with MIS-C. Increased levels of IFNγ-induced chemokines such as CXCL9, CXCL10, and CXCL11 were found to be associated with sJIA complicating macrophage activation syndrome [25]. Now, a trial of anti-IFNγ monoclonal antibody (emapalumab) in sJIA is on-going [26].”
Comment 6: Furthermore, the study limitations are not discussed properly. How about the subjectivity of ultrasound assessment? Have all patients been assessed by the same pediatric cardiologist or maybe two different physicians? If yes, did they perform a cross-check of their results? The study group is not only a small one, above all, but it is also a heterogeneous group that has been further subdivided. Moreover, the differences in the clinical presentation of MIS-C/PIMS between subsequent COVID-19 waves impede drawing consistent conclusions.
Response 6: Echocardiography of all patients was assessed by the same pediatric cardiologist who has more than five years of experience in pediatric cardiology. We clarified this part as follows:
“All patients underwent echocardiography by the same pediatric cardiologist (Vivid E9, GE Vingmed echocardiograph, General Electric, Horten, Norway).”
We agree with you and have expanded the section describing the study limitations.
"Our study was limited by its single-center design and small sample size. Furthermore, the clinical heterogeneity of the patient cohort may be due to successive waves of COVID-19 with variable severity as new SARS-Cov-2 variants emerged. The absence of a healthy control group is another limitation of this study. However, given the relatively low numbers of other studies reporting MIS-C, this report has added to the evidence base for the role of baseline CXCL10/IP10 measurement, where available, as a new biomarker for the prediction of LV dysfunction."
Comment 7: Additionally, the article is seriously linguistically flawed and requires thorough review by a native speaker. Many sentences have inappropriate syntax, probably adapted from another language. Furthermore, here are just a few examples of mistakes: "Increased levels of CXCL10/IP10 was" (line 36), "predictor potential" (line 72), "compare its power with those" (line 79).
Response 7: The English editing was performed by a native speaker of English with more than 25 years experience in medical research, writing, editing, refereeing and reviewing. We hope that the grammar and syntax is now acceptable to Reviewer 1.
“Acknowledgement: The authors are grateful to Mr. Jeremy Jones of the Academic Writing Department of Kocaeli University, Izmit, Turkey, for his assistance in editing
the English used and for his help and advice concerning the contents of this manuscript.”
Comment 8: I do believe that the revised version of the manuscript may enlighten future readers with its originality. I am looking forward to reading this paper after implementing of all suggested corrections.
Response 8: We are very grateful to expert Reviewer 1 for their kind and encouraging comments. We have endeavored to comply with all of your criticisms and suggestions and we also believe that the revision is a better article for it. If any further changes are needed, we would be happy to make a further revision.
Reviewer 2 Report
dear editors,
thanks for asking me to review the paper "CXCL10/IP10 as a biomarker linking multisystem hyperinflammatory syndrome and left ventricular dysfunction in children with SARS-CoV-2" by Basar and colleagues.
in this work, the authors present their findings regarding the potential predictive value of ifn gamma induced CXCL10 in MIS-C/kawasaki-like patients, compared with other inflammatory biomarkers.
even though other authors already published their data about this topic, sars-cov-2 infection and its related cytokines/chemokines alterations are still a debated topic and more insights are welcomed.
overall, the paper is well written, M&M are clear and the results are well presented.
my main concern regards the point where the authors conclude that CXCL10 may be a potential marker for LV dysfunction. It is my opinion that it must be stressed that this chemokine evaluation is available only in selected and experimental settings (i.e. not in all Ped E.D.), therefore i won't be so confident in suggesting its potential role in real-life situations. Moreover, hence the importance of this chemokine, i would like also to read the opinion of the authors regarding potential target therapies (emapalumab maybe?)
In addition: what is CXCL10 normal range? did you also used some healthy donors as control? i suggest to explain this in the text
Author Response
Reviewer 2:
Comment 1: Thanks for asking me to review the paper "CXCL10/IP10 as a biomarker linking multisystem hyperinflammatory syndrome and left ventricular dysfunction in children with SARS-CoV-2" by Basar and colleagues. In this work, the authors present their findings regarding the potential predictive value of ifn gamma induced CXCL10 in MIS-C/kawasaki-like patients, compared with other inflammatory biomarkers. Even though other authors already published their data about this topic, sars-cov-2 infection and its related cytokines/chemokines alterations are still a debated topic and more insights are welcomed. Overall, the paper is well written, M&M is clear and the results are well presented.
Response 1: We would like to express our heart-felt gratitude to expert Reviewer 2 for their very kind and encouraging comments. We agree that the evidence base of the relationship between SARS-Cov-2 infection, MIS-C and CXCL10/IP10 (and other chemokines) is limited and so hope that the addition of the data from our, admittedly heterogeneous, cohort of 36 patients is helpful to our colleagues, wherever they practice.
Comment 2: My main concern regards the point where the authors conclude that CXCL10 may be a potential marker for LV dysfunction. It is my opinion that it must be stressed that this chemokine evaluation is available only in selected and experimental settings (i.e. not in all Ped E.D.), therefore I won't be so confident in suggesting its potential role in real-life situations. Moreover, hence the importance of this chemokine, I would like also to read the opinion of the authors regarding potential target therapies (emapalumab maybe?)
Response 2: Thank you for your apposite comment. We agree that measurement of CXCL10/IP10 is not available to all. We have noted this in the last sentence of the Discussion " However, given the relatively low numbers of other studies reporting MIS-C, this report has added to the evidence base for the role of baseline CXCL10/IP10 measurement, where available, as a new biomarker for the prediction of LV dysfunction."
We also included some discussion of the possible role of emapalumab, as follows:
“Anti-IFNγ monoclonal antibody (emapalumab) may be an alternative treatment in MIS-C patients presenting with LV systolic dysfunction.”
Comment 3: In addition: what is CXCL10 normal range? did you also use some healthy donors as a control? I suggest explaining this in the text
Response 3: Unfortunately, there is no normal range for CXCL10. So we believed the absence of a healthy control group is a limitation. However, significant differences in the levels of CXCL10/IP10 between the three clinical groupings, and in particular, the patients with LV dysfunction may support our findings.
Reviewer 3 Report
The manuscript by Basar et al titled, “CXCL10/IP10 as a biomarker linking multisystem hyperinflammatory syndrome and left ventricular in children with SARS-CoV-2” have investigated diagnostic accuracy of CXCL10/IP10 for left ventricular 28 dysfunction in multisystemic hyperinflammatory syndrome (MIS-C). in pediatric patients. However, the content of this manuscript is not comprehensive, and lacks novelty.
Major comments:
- The introduction section lacks sufficient detail and is not focused enough why the study was taken which weakens the focus of the manuscript.
- In introduction section, Herein, we aim to evaluate the potential predictive value of CXCL10/IP10 in the course of MIS-C and compare its power with those of other inflammatory markers. How do authors think, this study is novel and with what other potential/known markers have compared the levels of CXCL10/IP10?. A case report by Baresi et al 2021: Analysis of Inflammatory Cytokines IL-6, CCL2/MCP1, CCL5/RANTES, CXCL9/MIG, and CXCL10/IP10 in a Cystic Fibrosis Patient Cohort During the First Wave of the COVID-19 Pandemic has already reported higher levels of CXCL10 in pediatric patients with SAR-CoV2.
- The study lacks proper controls for comparison such as authors should have collected serum from healthy individuals as well as patients with ventricular dysfunction for comparison.
- Methodology in the study should be provided in detail. The authors have used only ELISA to detect CXCL10/IP10 and should other techniques which are more sensitive such as liquid chromatography to detect the CXCL10/IP10 levels in pediatric patients.
- In conclusion section, CXCL10/IP10 plays a role in the pathophysiology of MIS-C related to SARS-COV2 and we suggest that it is a potential biomarker for cardiac involvement in these patients. The authors should support more evidence to come to this conclusion.
Minor comments:
- Abbreviations should be explained before using them in rest of the manuscript.
- The manuscript needs an English language editing and there are several spelling mistakes through the manuscript.
Author Response
Reviewer 3: The manuscript by Basar et al titled, “CXCL10/IP10 as a biomarker linking multisystem hyperinflammatory syndrome and left ventricular in children with SARS-CoV-2” has investigated the diagnostic accuracy of CXCL10/IP10 for left ventricular dysfunction in multisystemic hyperinflammatory syndrome (MIS-C). in pediatric patients. However, the content of this manuscript is not comprehensive and lacks novelty.
Response 1: We are sorry that Reviewer 3 considers our report unfavorably. We believe that, given the dearth of information on the clinical natural history of MIS-C and in particular the potential role for CXCL10/IP10 in the identification of pediatric Covid-19 patients who may suffer severe cardiac complications, our report, even with its limitations, can add to the evidence base available to all of us. We hope that by responding to the comments of Reviewer 3 and the other expert referees, our article has been improved and its limitations made more evident, so that our findings, despite the limitations, may help to inform and shape future research.
Comment 2: The introduction section lacks sufficient detail and is not focused enough on why the study was taken which weakens the focus of the manuscript. In the introduction section, Herein, we aim to evaluate the potential predictive value of CXCL10/IP10 in the course of MIS-C and compare its power with those of other inflammatory markers. How do authors think, this study is novel and with what other potential/known markers have compared the levels of CXCL10/IP10?. A case report by Baresi et al 2021: Analysis of Inflammatory Cytokines IL-6, CCL2/MCP1, CCL5/RANTES, CXCL9/MIG, and CXCL10/IP10 in a Cystic Fibrosis Patient Cohort During the First Wave of the COVID-19 Pandemic has already reported higher levels of CXCL10 in pediatric patients with SAR-CoV2.
Response 2: We revised the background aims, as stated in the Introduction, as follows:
“Loré et al. [7] evaluated 53 potential biomarkers to determine the factors influencing outcome in COVID-19 and they found that CXCL10/IP10 was the best predictive biomarker for outcome in adults with COVID-19. However, the relationship between CXCL10/IP10 levels and disease course in MIS-C patients has not been clarified fully. A study comparing pediatric and adult COVID-19 patients showed increased and similar IFN gene responses in both groups while this antiviral response resolves faster in children [8]. Caldarale et al. [9] demonstrated increased levels of IL-6, CCL2, CXCL8, CXCL9, and CXCL10/IP10 in patients with MIS-C. However, differences in CXCL10/IP10 levels among clinical subgroups have not been investigated. Herein, we aim to evaluate whether CXCL10/IP10 levels change or not according to the clinical course of MIS-C patients and we also compare CXCL10/IP10 levels with those of other inflammatory markers.”
Comment 3: The study lacks proper controls for comparison such as authors should have collected serum from healthy individuals as well as patients with ventricular dysfunction for comparison.
Response 3: We have highlighted the lack of a healthy control group as a limitation of our study. However, showing the significantly elevated CXCL-10 levels in patients with LV dysfunction, compared to the other MIS-C subgroups may support our findings. We have rewritten the "limitations" paragraph of the Discussion:
"Our study was limited by its single-center design and small sample size. Furthermore, the clinical heterogeneity of the patient cohort may be due to successive waves of COVID-19 with variable severity as new SARS-Cov-2 variants emerged. The absence of a healthy control group is another limitation of this study. However, given the relatively low numbers of other studies reporting MIS-C, this report has added to the evidence base for the role of baseline CXCL10/IP10 measurement, where available, as a new biomarker for the prediction of LV dysfunction."
Comment 4: Methodology in the study should be provided in detail. The authors have used only ELISA to detect CXCL10/IP10 and should other techniques which are more sensitive such as liquid chromatography to detect the CXCL10/IP10 levels in pediatric patients.
Response 4: Once again, Reviewer 3 is correct but for consistency of methodology and ease of testing we chose to use an ELISA technique for the measurement of CXCL10/IP10. This approach has been used by many other published studies, so we are confident that it is appropriate in this study too.
Comment 5: In the conclusion section, CXCL10/IP10 plays a role in the pathophysiology of MIS-C related to SARS-COV2 and we suggest that it is a potential biomarker for cardiac involvement in these patients. The authors should support more evidence to come to this conclusion.
Response 5: We thank Reviewer 3 for this observation and agree. WE have therefore rewritten the Conclusion, as follows:
“In conclusion, we suggest that CXCL10/IP10 plays a role in the pathophysiology of MIS-C related to SARS-COV2. Evaluating the accuracy and utility of CXCL10/IP10 in larger prospective studies may help clarify this hypothesis and the exact etiopathology of MIS-C. If CXCL10/IP10 has a central role in this, then novel targeted therapies may emerge, such as anti-IFNγ monoclonal antibodies.”
Comment 6: Abbreviations should be explained before using them in the rest of the manuscript. The manuscript needs English language editing and there are several spelling mistakes throughout the manuscript.
Response 6: Abbreviations were written in full at first use and then used consistently thereafter.
The English used in the Revision has been reviewed and edited by an experienced medical writer, editor, referee and reviewer with more than 25 years experience in the field. We have added an acknowledgement to him at the end of the revision.
Round 2
Reviewer 1 Report
I would like to thank the authors for the effort made to implement all of my suggestions, I am satisfied with the revised version of the manuscript.
Reviewer 3 Report
The authors response to most of the comments has significantly improved the manuscript.